# Bioassay-Guided Fractionation of *Pittosporum angustifolium* and *Terminalia ferdinandiana* with Liquid Chromatography Mass Spectroscopy and Gas Chromatography Mass Spectroscopy Exploratory Study

**DOI:** 10.3390/plants13060807

**Published:** 2024-03-12

**Authors:** Janice Mani, Joel Johnson, Holly Hosking, Luke Schmidt, Ryan Batley, Ryan du Preez, Daniel Broszczak, Kerry Walsh, Paul Neilsen, Mani Naiker

**Affiliations:** 1College of Science and Sustainability, CQUniversity, North Rockhampton, QLD 4701, Australia; joel.johnson@cqumail.com (J.J.); h.hosking2@cqu.edu.au (H.H.); ryan.batley@cqumail.com (R.B.); r.dupreez@cqu.edu.au (R.d.P.); k.walsh@cqu.edu.au (K.W.); p.neilsen@cqu.edu.au (P.N.); m.naiker@cqu.edu.au (M.N.); 2Institute for Future Farming Systems, CQUniversity, Bundaberg, QLD 4670, Australia; 3School of Biomedical Sciences, Queensland University of Technology, Brisbane, QLD 4000, Australia; luke.schmidt@hdr.qut.edu.au (L.S.); daniel.broszczak@qut.edu.au (D.B.); 4Jawun Research Centre, Cairns, QLD 4870, Australia

**Keywords:** bioassay-guided fractionation, *Pittosporum angustifolium*, bioactive compounds, phytochemicals, phenolic compounds, antioxidants, LC-MS/MS

## Abstract

Bioprospecting native Australian plants offers the potential discovery of latent and novel bioactive compounds. The promising cytotoxic and antibacterial activity of methanolic extracts of *Pittosporum angustifolium* and *Terminalia ferdinandiana* led to further fractionation and isolation using our laboratory’s bioassay-guided fractionation protocol. Hence, the aim of this study was to further evaluate the bioactivity of the fractions and subfractions and characterize bioactive compounds using liquid chromatography mass spectroscopy (LC-MS/MS) and gas chromatography MS (GC-MS). Compounds tentatively identified in *P. angustifolium* Fraction 1 using LC-ESI-QTOF-MS/MS were chlorogenic acid and/or neochlorogenic acid, bergapten, berberine, 8′-epitanegool and rosmarinic acid. GC-MS analysis data showed the presence of around 100 compounds, mainly comprising carboxylic acids, sugars, sugar alcohols, amino acids and monoalkylglycerols. Furthermore, the fractions obtained from *T. ferdinandiana* flesh extracts showed no cytotoxicity, except against HT29 cell lines, and only Fraction 2 exhibited some antibacterial activity. The reduced bioactivity observed in the *T. ferdinandiana* fractions could be attributed to the potential loss of synergy as compounds become separated within the fractions. As a result, the further fractionation and separation of compounds in these samples was not pursued. However, additional dose-dependent studies are warranted to validate the bioactivity of *T. ferdinandiana* flesh fractions, particularly since this is an understudied species. Moreover, LC-MS/GC-MS studies confirm the presence of bioactive compounds in *P. angustifolium* Fraction 1/subfractions, which helps to explain the significant acute anticancer activity of this plant. The screening process designed in this study has the potential to pave the way for developing scientifically validated phytochemical/bioactivity information on ethnomedicinal plants, thereby facilitating further bioprospecting efforts and supporting the discovery of novel drugs in modern medicine.

## 1. Introduction

Bioprospecting involves a multidisciplinary approach that entails the systematic discovery, isolation and identification of new bioactive molecules from natural biological reserves, such as plants. Whilst medicinal plants have been used to treat various health conditions for centuries, conventional bioprospecting methods are often time-consuming and expensive [1]. Moreover, the complexity of plant matrices and the occurrence of numerous phytochemicals can make the separation and analysis process quite challenging.

The process of separating phytochemicals involves isolating the constituents in the plant extracts or effective parts and purifying them into monomer compounds using physical and chemical methods [2]. Conventional isolation methods include solvent extraction, precipitation, crystallization, fractional distillation, salting out and dialysis. More modern techniques include column chromatography, high-performance liquid chromatography, ultrafiltration and high-performance liquid droplet counter current chromatography. There are several chromatographic techniques available for the identification and quantification of phenolic compounds in plants, including thin-layer chromatography (TLC), gas chromatography (GC), high-performance liquid chromatography alone or coupled with mass spectrometry (MS), capillary electrophoresis (CE) and micellar electro-kinetic chromatography (MEKC) [3].

Phenolic compounds are a highly complex class of naturally occurring molecules that possess a range of therapeutic properties. As a result, significant interest has been devoted to their analysis in medicinal plants and food samples. High-performance liquid chromatography (HPLC) is the most commonly utilized separation technique for this purpose [4,5,6,7].

High-performance liquid chromatography equipped with a fraction collector was the analytical tool of choice for the separation and isolation of fractions/subfractions of extracts in this study due to its characteristic features of high efficiency, speed, and automation. The separation of compounds is based on the principles of the adsorptive capacity of the column stationary phase to different compounds, the molecular size of the compounds, the difference in dissociation degrees of the chemical constituents and the difference in partition coefficients between the stationary and mobile phase [2].

The medicinal properties of Australian native plants due to their strong expression of special metabolites such as phenolics, alkaloids, sterols, essential oils and liginans [8] are indictive of the potential discovery of novel prophylatic and/or therapeutic phytochemicals. *Pittosporum angustrifolium* Lodd, commonly known as Gumbi gumbi, is an Australian native shrub or tree species from the Pittoporaceae family [9]. It is as a popular traditional medicine which was first discovered and used by Indigenous Australians for treating pathogenic diseases [9] and has been shown to have anticancer properties; hence, it is sometimes referred to as “Queensland’s anticancer tree” [10]. *Terninalia ferdinandiana* Exell, commonly known as Kakadu plum, is a moderately sized, semi-deciduous native Australian tree that bears fruit upon reaching maturity [11]. This fruit has been reported to contain the highest amount of ascorbic acid, with levels as high as 6% of recorded wet weight, which is estimated to be 900 times higher than that of blueberry (g g^−1^) [12,13].

Based on the findings from our previous study [14], the objective of this study was to further fractionate Fraction 1 of *P. angustifolium* methanolic crude extracts into subfractions, utilizing phase 3 of the proposed bioassay-guided fractionation protocol design (Figure 1), and to perform LC-MS and GC-MS analysis for compound characterization. This is among the first/few studies investigating the bioactivity of fractions from *P. angustifolium.* Additionally, since *T. ferdinandiana* flesh lyphophized extracts have also shown some therapeutic potential, fractionation and bioassay testing on these fractions were also included.

## 2. Results and Discussion

### 2.1. Pittosporum angustifolium Fractions and Subfractions

#### 2.1.1. LC-MS Analysis of *P. angustifolium* Fraction 1

While the subfractions of *P. angustifolium* Fraction 1 did not demonstrate significant cytotoxic properties or contain compounds of interest, Fraction 1 was found to be effective in reducing cell viability in the tested cancer cell lines and had a higher selective index, suggesting its effectiveness in killing cancer cells while minimizing harm to normal cells compared to the other fractions [14]. Moreover, given the possibility of a synergistic effect at play in dictating the cytotoxic behavior of *P. angustifolium* Fraction 1, characterizing the potential phenolic metabolites in the fraction was considered a valuable pursuit.

Using LC-ESI-QTOF-MS/MS, the untargeted screening and characterization of phenolic compounds in Fraction 1 was performed. The obtained MS/MS spectra were compared with NIST and PubMed database libraries, as well as the published literature, to putatively confirm the presence of phenolic metabolites (Table 1). Among ten prominent peaks, only peaks 1, 4 and 7–9 were tentatively identified (Figure 2) in Fraction 1.

Compound **1** (*m*/*z* 355.10) mass spectrum data, as presented in Table 1, were identified as either chlorogenic acid or its isomer, neochlorogenic acid, both of which belong to the caffeoylquinic acid class of molecules. These compounds are known for their strong antioxidant, anticancer, anti-inflammatory, and antifungal properties [16]. Previous studies have also identified chlorogenic acid in *P. angustifolium* leaves [17,18,19]. Additionally, our previous work has suggested the presence of chlorogenic acid in crude MeOH extracts of *P. angustifolium* leaves [14,20].

Compound **4** (*m*/*z* 336.20) mass spectrum data, as presented in Table 1, were tentatively identified as berberine (2,3-methylenedioxy-9,10-dimethoxyprotoberberine chloride), a benzyl tetra isoquinoline alkaloid. Berberine been previously extracted from roots of various plants such as *Berberis vulgaris*, *B. aristotle*, *B. aquifolium*, *Hydrastus canadensis*, *Pellodendron chenins* and *Coptidis rhizomes* [21]. Numerous authors have reported the broad-spectrum therapeutic potential of berberine due to its action against diabetes, hypertension, depression, obesity, inflammation and cancer [21,22,23,24]. However, this is the first study to tentatively propose the occurrence of berberine in *P. angustifolium*. Therefore, further investigation to confirm this finding is warranted.

Compound **7** (*m*/*z* 217.10) mass spectrum data, as presented in Table 1, were identified as bergapten (5-methoxypsoralen), which belongs to the class furocoumarin. This furanocoumarin derivate is commonly found in bergamot essential oil, other citrus essential oils and grapefruit juice, as well as in a wide variety of medicinal plants from the *Rutaceae* and *Umbelliferae* families such as figs, parsley, celery and anise [25]. Pharmacological studies have shown that bergapten has various properties, including neuroprotection, organ protection, anticancer, anti-inflammatory, antimicrobial and antidiabetic effects [25,26]. However, this is the first study to report its potential occurrence in *P. angustifolium*, and further investigations are required to confirm this claim.

Compound **8** (*m*/*z* 361.20) mass spectrum data, as presented in Table 1, were tentatively identified as rosmarinic acid, which belongs to the hydroxycinnamic acid class of phenolic acids and is commonly found in fruits, herbs and medicinal plants (Ali et al., 2022 [27]). It is produced by *Boraginaeceae* and the *Nepetoideae* subfamily of the *Lamiaceae* plant species. Initially, it was extracted as a pure compound from rosemary (*Rosmarinus officinalis*) [28]. This compound has shown potent biological activities in combating human diseases such as cancer, diabetes, neurodegenerative disorders, cardiovascular disease and inflammatory disorders [29]. While this is the first study to tentatively report its occurrence in *P. angustifolium*, further investigation is needed to confirm this finding.

Compound **9** (*m*/*z* 403.2) mass spectrum data, as presented in Table 1, were tentatively identified as 8′-epitanegool, classified as phenylpropanoids. A previous study demonstrated promising in silico antiviral results similar to the main alkaloids [30]. This compound has been previously identified in *Tinospora sinensis*, a type of Chinese folk medicine [16], and the literature on its therapeutic potential is limited or non-existent. Additionally, this is the first study to tentatively identify this compound in *P. angustifolium*.

If the identities of these compounds are confirmed in Fraction 1 of *P. angustifolium*, then it will not only strongly support its antioxidant and anticancer properties, as determined in this study, but also the anecdotal claims of Indigenous Australians [18]. Furthermore, detailed studies to confirm the identity of these compounds and/or to discover other bioactive compounds in *P. angustifolium* crude, fractions and subfractions were conducted using GC-MS.

#### 2.1.2. GC-MS Analysis of *Pittosporum angustifolium* Fraction 1 and Subfractions

Targeted GC-MS/MS analysis in MRM acquisition mode identified a total of 103 compounds, belonging to various classes of primary and secondary plant metabolites. The main primary metabolites identified are classified as carbohydrates, amino acids, proteins, lipids, purines and pyrimidines of nucleic acids. In addition, the secondary metabolites identified were classified into the following three main groups: (a) nitrogen-containing compounds such as alkaloids, glucosinolates and cyanogenic glycosides; (b) phenolic compounds such as phenylpropanoids and flavonoids; and (c) terpenes [31]. However, only twenty predominantly found compounds in Fraction 1 and Fraction 1 subfractions are reported in Figure 3 and Table 2, which mainly belong to the class of carbohydrates and amino acids.

Inositol (Figure 4), the third most abundant compound identified in Fraction 1, was of interest due to its previously reported anti-atherogenic, anti-oxidative, anti-inflammatory and anticancer properties [32]. Clinical trials using inositol in pharmacological doses have shown promising results in the management of gynecological diseases, respiratory stress syndrome, Alzheimer’s disease, metabolic syndrome and cancer [33]. Inositol occurs naturally in all eukaryotes and is involved in several biological processes [32]. In mammals, inositol is produced in the liver and kidney, and myo-inositol (inositol isomer) and its derivatives in particular are involved in biological functions which include the modulation of glucose metabolism, calcium release in cell signaling, chromatin and CSK remodeling, gene transcription, proliferation, apoptosis and proper structural development [33]. In plants, inositol is well known for acting a stress ameliorator and controls multiple aspects of plant signaling and physiology [34]. On this premise, the cytotoxic effects of Fraction 1 may likely be due to the predominant occurrence of inositol. However, to our knowledge, this study is the first to report the occurrence of inositol in *P. angustifolium*, and thus, further investigations are required to confirm this finding.

#### 2.1.3. Cytotoxic Activity and HPLC Profiling

The subfractionated products obtained from *P. angustifolium* Fraction 1 (GGLX F1 S1-3) were subjected to an anticancer bioassay, and the results are presented in Figure 5. There was no significant difference (*p* > 0.05) in the percentage cell viability between of the treated cells and negative control for all the tested cell lines, except for the HT29 cells (*p* < 0.05), where all three subfractions demonstrated some cytotoxic activity. The low bioactivity observed in the subfractions may be attributed to the low doses of compounds present in these subfractions, as indicated by low peak signals in the HPLC chromatograms shown in Figure 6. Additionally, the cytotoxic effect observed in the methanolic extracts of *P. angustifolium* and its fractions may be due to the synergistic action of different compounds. On the other hand, the subfractions could contain fewer isolated compounds, which may explain the low or no cytotoxicity observed.

Although numerous studies have identified triterpenoid saponins, terpenoids, phenols, and coumarin compounds isolated from *P. angustifolium* as having potential anticancer properties [10,17,18,35,36], only a few have been shown to be effective in vitro. Backer et al. (2016) screened ten acylated saponins for their ability to inhibit human DNA-topoisomerase I, an enzyme responsible for resolving torsional stress associated with DNA replication, transcription, and chromatin condensation [37]. Inhibitors of DNA-topoisomerase I can inhibit the proliferation of cancer cells, and such agents are commonly used in chemotherapy for their antiproliferative effects. However, their effects on the metastasis of cancer cells remain unclear [38].

In previous work, Backer et al. (2015) isolated two new taraxastane-type triterpene saponins, which were evaluated against four cell lines [9]. However, no cytotoxic activity was observed up to a concentration of 130 µM. In a similar subfractionation study of *Syzygium polyanthum* (Wight.), the crude methanol extract showed higher bioactivity in terms of hypoglycemic effect compared to the fraction, subfraction, and squalene (the major chemical compound and a triterpene isolated from *S. polyanthum* leaf extract) [39]. Therefore, the cytotoxic properties of *P. angustifolium* could possibly be due to a synergistic effect, similar to that seen in other studies.

In addition, authentic standards of selected polyphenols (4-hydroxybenzoic acid, caffeic acid, catechin, catechol, chlorogenic acid, gallic acid, isovanillic acid, neochlorogenic acid, protocatechuic acid, syringic acid, tyrosol and vanillic acid) were subjected to the same HPLC gradient elution as the subfractions, and the combined chromatograms with retention times are shown in Figure 7. When compared to the retention times of the predominant peaks in the chromatograms of the subfractions (Figure 6), only one peak at a retention time of 4.412 min in Subfraction 1 could tentatively be identified as gallic acid based on it having a similar UV spectrum to the standard. However, there were some discrepancies between the retention time of the standard gallic acid (4.009 min) and the subfraction (4.412 min). In Subfraction 2, none of the retention time peaks or UV spectra matched any of the authentic standards. On the other hand, the retention time of a predominant peak at 7.768 min in Subfraction 3 was tentatively identified as 4-hydroxybenzoic acid, as a similar retention time (7.775 min) and UV spectrum was evident (Figure 7).

As the subfractions of *P. angustifolium* did not show any significant cytotoxic activity, further separation and isolation was deemed impractical and hence was not pursued.

### 2.2. Terminalia ferdinandiana Fractions

#### 2.2.1. Cytotoxic Activity

The lyophilized methanolic flesh extract of *T. ferdinandiana* was fractionated into four fractions, as listed in Table 3. The cytotoxicity of these fractions was evaluated against HeLa, HT29 and PH5CH8 cell lines using the concentrations given in Table 3. The results of this experiment are presented in Figure 8.

None of the fractions showed significant (*p* > 0.05) cytotoxic activity against HeLa and PH5CH8 cell lines, except for slight cytotoxicity observed against the HT29 cell line. This cytotoxicity was significantly different from the negative control (*p* < 0.05). To minimize the matrix effect, lower doses of the fractions were used in the bioassay. However, in future studies, higher doses could be utilized to investigate potential higher toxicity.

While the previous literature [40,41,42] has shown cytotoxic properties of *T. ferdinandiana*, there are limited studies on their fractions [43]. As such, this study is crucial in paving the way for future fractionation studies of this species.

#### 2.2.2. Antibacterial Activity

Considering the promising antibacterial activity demonstrated by the crude flesh extract of *T. ferdinandiana* [14], the fractions were also evaluated against four bacterial strains. The results are presented in Table 4.

The antibacterial activity exhibited by the fractions was relatively mild, and in some cases (Fraction 3), no activity was observed. Fraction 2 demonstrated the highest activity and was the only fraction that inhibited the growth of all tested bacterial strains compared to the other fractions, possibly because it was the most concentrated fraction. Overall, even though the flesh extracts of *T. ferdinandiana* fruit have previously demonstrated antibacterial activity in several studies [11,44,45], including our previous studies [14], the low activity of the fractions suggests that the bioactive compounds in the lyophilized extract may work synergistically to produce its antibacterial property.

Despite the small amount of existing literature on the therapeutic potential of *T. ferdinandiana*, this study provides the first report on the bioactivity of its fractions. However, further rigorous testing is necessary to validate the data obtained, especially in terms of dose-dependent anticancer and antibacterial effects. Therefore, more elaborate investigation into this understudied native fruit is warranted.

## 3. Materials and Methods

### 3.1. Reagents

Hydrochloric acid and sodium carbonate were purchased from Chem-Supply (Gillman, Australia). All other reagents, including the HPLC-grade methanol, were purchased from Sigma-Aldrich (Melbourne, Australia). Some reagents used in the cytotoxicity analysis, which included the CellTiter 96^®^ AQueous Assay (composed of solutions of tetrazolium compound [3-(4,5-dimethylthiazol-2-yl)−5-(3-carboxymethoxyphenyl)−2-(4-sulfophenyl)−2H-tetrazolium, inner salt; MTS(a)]) and an electron-coupling reagent (phenazine methosulfate; PMS), commonly known as MTS reagent, and fetal bovine serum (FBS) were obtained from Promega (Madison, WI, USA) and Scientifix (Clayton, Australia), respectively. Dulbecco’s Modified Eagle’s Medium—high glucose (DMEM) and Dulbecco’s Phosphate Buffered Saline (DPBS) solution were kept in the dark at 4 °C, while the other reagents used in the bioassays were frozen until they were required for use. All dilutions and assay preparations used Milli-Q water. All reagents used were of analytical-grade or higher purity.

### 3.2. Sample Extraction

Approximately 2.5 g of powdered plant material (leaves of *P. angustifolium* and fruit of *T. ferdianadiana*) was extracted in 75 mL of 90% methanol as previously detailed [14]. The supernatant obtained was filtered using 0.45 µm Advantec filter paper and evaporated under reduced pressure at 27 °C to a semi-solid consistency using a rotary evaporator. The semi-solid product was redissolved in approximately 25 mL of Milli-Q water and freeze-dried under vacuum (Flexi-Dry Freeze-dryer, −47 °C, 277 mTorr) for 72 h to obtain a fine lyophilized product, which was stored at 4 °C in the dark until required.

### 3.3. HPLC Fractionation and Subfractionation of P. angustifolium Extract

The HPLC conditions mentioned in our previous publication [14] were followed with slight modifications. Briefly, a reversed-phase C18 column (Agilent Eclipse XDB-C18; 150 × 4.6 mm; 5 µm pore size) and guard cartridge (Gemini C18 4 × 2 mm) with an injection volume of 30 µL and a run time of 50 min with post run time of 5 min was used for column flushing. The time slicing feature of the Agilent fraction collector was used to collect only Fraction 1 (0 to 12 min) from 20 mg mL^−1^ of *P. angustifolium* lyophilized product. The volume collected after multiple runs was then rotary evaporated to a semi-solid consistency and reconstituted in 30 mL of Milli-Q water. This was then placed at −80 °C overnight and then freeze-dried for 72 h. A crystalline product with a mass of 87.9 mg was obtained. The HPLC chromatogram of *P. angustifolium* Fraction 1 is depicted in Figure 9.

The crystalline product obtained from Fraction 1 was redissolved in Milli-Q water at a concentration of 43.95 mg·mL^−1^ and subjected to HPLC fractionation using gradient elution, as described in Figure 9B, and an injection volume of 30 µL. Retention time zones showing predominant peaks were selected for time slicing, and fractions were collected from 0–3 min (Subfraction 1), 3–6 min (Subfraction 2) and 6–11 min (Subfraction 3), as depicted in Figure 9C.

### 3.4. HPLC Fractionation of T. ferdinandiana Fruit Flesh Extract

The same HPLC conditions as described above (Section 3.3) and in our previous publication [14] were followed with slight modifications to the gradient elution and injection volume. The gradient elution described in Figure 9B was used, and a sample injection volume of 30 µL was applied. A total run time of 50 min was allowed to ensure that all eluents were captured in the chromatogram, and a post run time of 10 min was allowed for column flushing.

Retention times showing predominant peaks were selected for time slicing and the collection of fractions from 0–6 min (Fraction 1), 6–16 min (Fraction 2), 16–30 min (Fraction 3) and 30–40 min (Fraction 4), as depicted in Figure 10.

### 3.5. Cytotoxicity Assay

The cytotoxicity of the subfractions of *P. angustifolium* Fraction 1 and fractions of *T. ferdinandiana* were assessed against HeLa (human cervical carcinoma), HT29 (human colorectal carcinoma), HuH7 (human liver carcinoma) and PH5CH8 (human epithelial non-neoplastic hepatocyte cell), obtained from the University of Adelaide, using the MTS assay previously described [14]. However, it was observed that the proliferation and general health of the HuH7 cells were compromised, and they were nonviable for use in the cell culture assay of *T. ferdinandiana* fractions in the later trials.

### 3.6. Antimicrobial Activity

The antimicrobial activity of *T. ferdinandiana* Fractions (1–4) were tested against the four bacterial strains (Gram positive—*Staphylococcus aureus* and Gram negative—*Escherichia coli*, *Salmonella typhi* and *Pseudomonas aeruginosa*) following the disk diffusion method [46] with slight modifications, as described previously [14].

### 3.7. LC-MS/MS Analysis of P. angustifolium Fraction

The methanolic extract of *P. angustifolium* was analyzed for targeted phenolic compounds using liquid chromatography tandem mass spectroscopy (LC-MS/MS). The analysis was performed using a Nexera X2 chromatography system, which was coupled with a Shimadzu LCMS-8040 system comprising a CBM-20A communications bus module, a DGU-20A5R degassing unit, LC-30AD pumps, an SIL-30AC autosampler and a CTO-20AC column oven. The analytical method used a Raptor biphenyl column (100 mm × 2.1 mm, 2.7 µm), 5 µL injection volume, 40 °C column temperature and a flow rate of 0.6 mL min^−1^. The mobile phase comprised water (phase A) and methanol (phase B), each containing 5 mM ammonium formate and 0.1% formic acid. The eluent was directed to the electrospray ionization (ESI) module.

A Shimadzu LCMS-8040 model triple-quadrupole mass spectrometer, equipped with an electrospray ionization (ESI) source, was used to perform targeted tandem mass spectrometry on the eluting compounds. Both positive and negative ionization modes were used depending on the ionization characteristics of each analyte. The ESI conditions used were an interface temperature of 350 °C, a DL (dissolution line) temperature of 250 °C and 400 °C in the ESI source. Nitrogen was used as the nebulizing gas and drying gas at flow rates of 3 L min^−1^ and 15 L min^−1^, respectively, and the interface voltage used was 4.50 kV. The LC-MS/MS data were collected and analyzed in LabSolutions software version 5.99 SP2 (Shimadzu, Kyoto, Japan). The PubChem database and offline version (accessed 21 March 2023) of the National Institute of Standards and Technology (NIST) Library was used to match the MS/MS spectra of phenolic compounds in the *P. angustifolium* extract.

### 3.8. GC-MS/MS Analysis of P. angustifolium Fraction 1 and Subfractions

All four samples were dissolved in 1 mL of chilled MS-grade water and mixed via vortex and kept on ice throughout processing where possible. Twelve aliquots were created for each sample. Three aliquots each of 1 μL, 10 μL, 100 μL and 200 μL were generated by transferring the sample into glass vial inserts. The aliquots were then frozen prior to lyophilization at 1 mbar (room temperature) for 3.5 h until dry. The sample aliquots were then sealed in auto-sampler vials and stored at −20 °C until analysis.

Samples were analyzed on a Shimadzu TQ8050 NX system following automated trimethylsilyl (TMS) derivatization using an AOC 6000 plus auto-sampler. The derivatization was accomplished by adding 25 μL of 30 mg mL^−1^ methoxyamine hydrochloride in pyridine to 10 μL of dried metabolite extract. The samples were then incubated at 37 °C for 2 h with continuous agitation. Following this, 25 μL of N,O-Bis(trimethylsilyl)trifluoroacetamide (BSTFA) was added, with further incubation at 37 °C for 1 hr with continuous agitation. The derivatized sample was then incubated at room temperature for 1 hr prior to injection. One microliter of the derivatized samples was injected onto the GC-MS/MS by the auto-sampler at a split ratio of 5:1 and a constant flow of 1.10 mL min^−1^, and the oven temperature was maintained at 100 °C. The samples were analyzed in Multiple Reaction Monitoring (MRM) mode using the Shimadzu Smart Metabolite Database containing 521 MRM metabolite targets. Prior to each sample being analyzed, four hexane blanks were analyzed to ensure there was no carry over between samples.

Data produced from the analysis of samples were extracted using LabSolutions Insight software 4.25.39636.3. Metabolites that were within ± 0.1 min of the predicted retention time and within ±25% of the relative ion ratio had their area under the curve reported for further statistical analysis. Where metabolites did not need these criteria, the area under the curve was not reported.

### 3.9. Statistical Analysis

All data were presented as means ± SEM for triplicate samples. Statistical analyses were performed using RStudio version 2022.12.0-353 software, with a one-way analysis of variance to determine the significance between the control and treated groups. A value of *p* < 0.05 was considered statistically significant.

## 4. Conclusions

Whilst *P. angustifolium* crude leaf extracts and fractions demonstrated strong cytotoxic activities, no significant activity was evident in the subfractions, thereby suggesting that the bioactivity may be attributed to the synergistic effect of the phenolic compounds present. Further testing of subfraction combinations may confirm the possibilities of compound synergy. Moreover, LC-MS/GC-MS studies confirm the presence of bioactive compounds in *P. angustifolium* Fraction 1/subfractions, which helps to explain the significant acute anticancer activity of this plant. Compounds tentatively identified in *P. angustifolium* Fraction 1 using LC-ESI-QTOF-MS/MS were chlorogenic acid and/or neochlorogenic acid, bergapten, berberine, 8′-epitanegool and rosmarinic acid. GC-MS analysis indicated the predominant occurrence of compound inositol in *P. angustifolium* Fraction 1, which may be responsible for its anticancer properties. This is the first study to report the occurrence of the above-mentioned compounds in *P. angustifolium*, and further investigations are required to confirm these findings. Furthermore, the fractions of *T. ferdinandiana* flesh extracts showed no cytotoxicity, except against HT29 cell lines, and only Fraction 2 showed some antibacterial activity against the bacterial strains tested. The reduced bioactivity in the *T. ferdinandiana* fractions may again also be due to loss of synergy as compounds become separated in the fractions. Further dose-dependent studies to validate the bioactivity of *T. ferdinandiana* flesh fractions is warranted for this understudied species. Overall, this study successfully applied the third phase of our developed bioassay-guided fractionation protocol (Figure 1). However, further studies and retests are warranted to confirm and check for the reproducibility of the results obtained.

## Figures and Tables

**Figure 1 plants-13-00807-f001:**
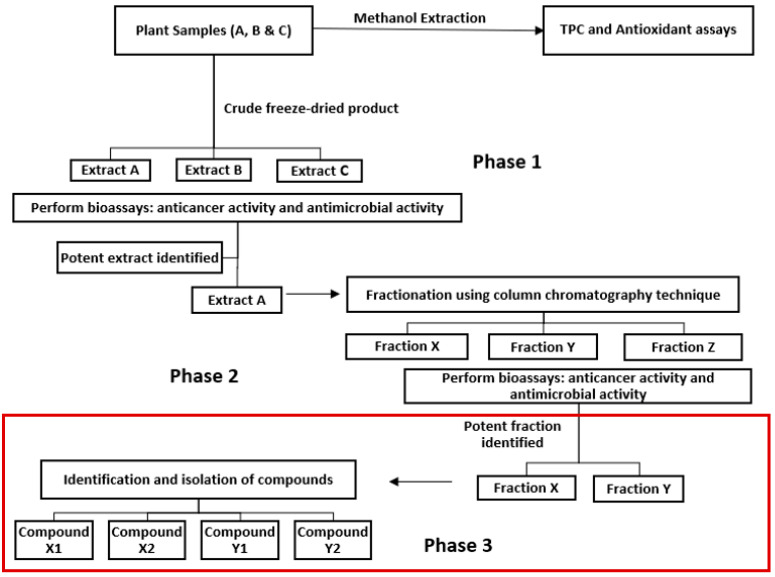
Bioassay-guided fractionation protocol design [14].

**Figure 2 plants-13-00807-f002:**
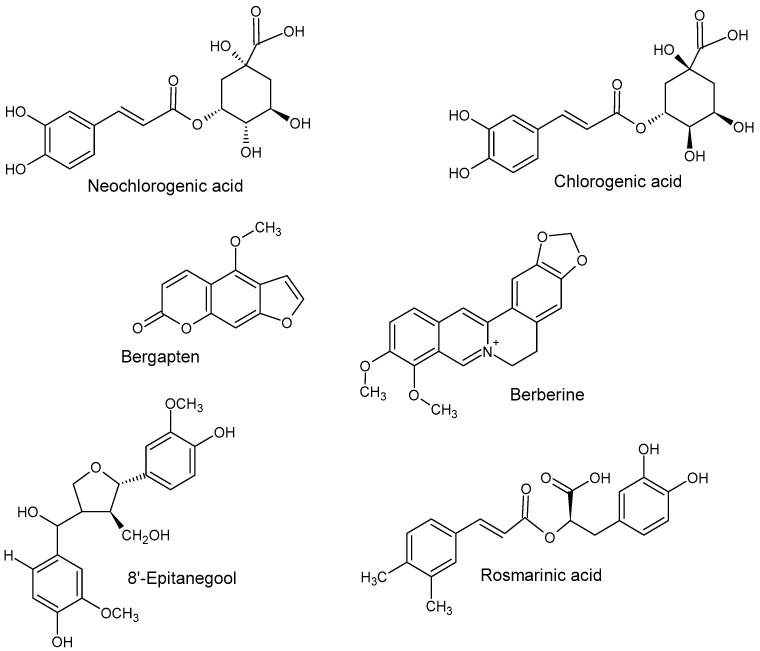
Proposed phenolic metabolites in *Pittosporum angustifolium* Fraction 1.

**Figure 3 plants-13-00807-f003:**
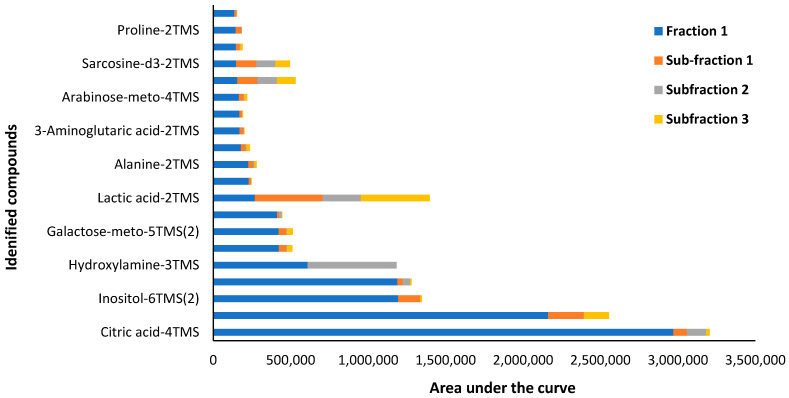
Targeted GC-MS/MS compounds identified in *Pittosporum angustifolium* Fraction 1 and Fraction 1 Subfractions 1, 2 and 3.

**Figure 4 plants-13-00807-f004:**
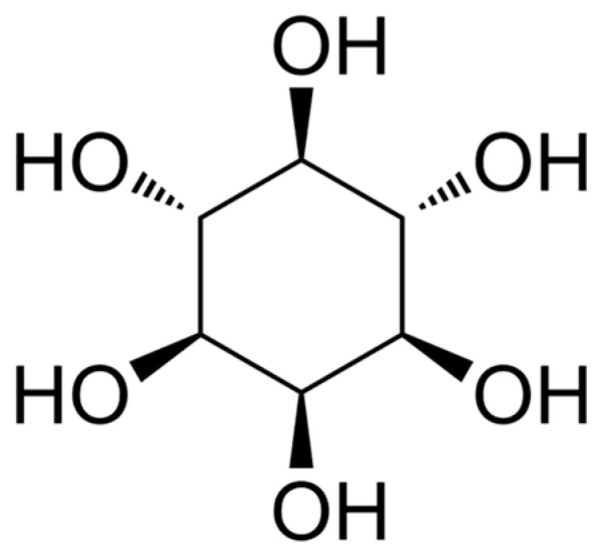
Chemical structure of inositol.

**Figure 5 plants-13-00807-f005:**
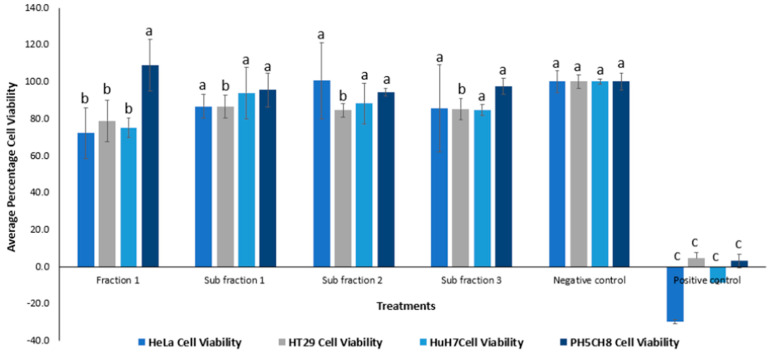
Percentage cell viability of cell lines treated with *Pittosporum angustifolium* Fraction 1 of concentration 5340 µg mL^−1^ and Fraction 1 Subfractions 1, 2 and 3 with concentrations of 16.55, 12.15 and 16.15 µg mL^−1^, respectively. One-way ANOVA test indicated no significant difference (*p*-value > 0.05) in cytotoxicity between the different subfractions for the same cell line, denoted by the same letters on the respective bars, except in the case of HT29 cells. Negative control: cells without treatment, positive control: cell treated with 50 ug mL^−1^ cisplatin (chemotherapy drug).

**Figure 6 plants-13-00807-f006:**
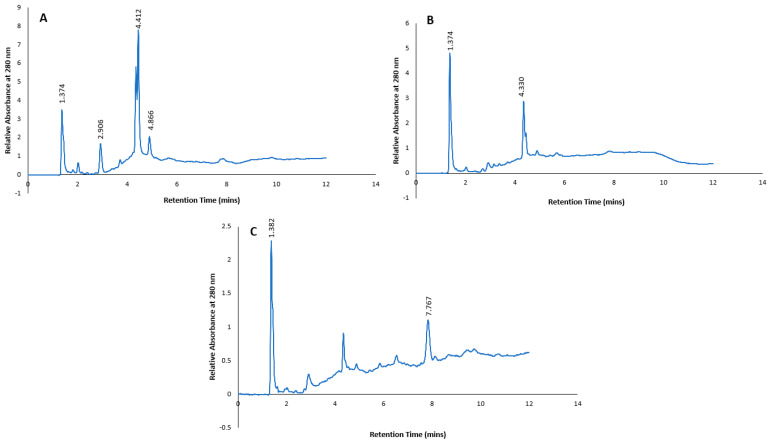
HPLC profiles of *Pittosporum angustifolium* Subfraction 1 (**A**), Subfraction 2 (**B**) and Subfraction 3 (**C**).

**Figure 7 plants-13-00807-f007:**
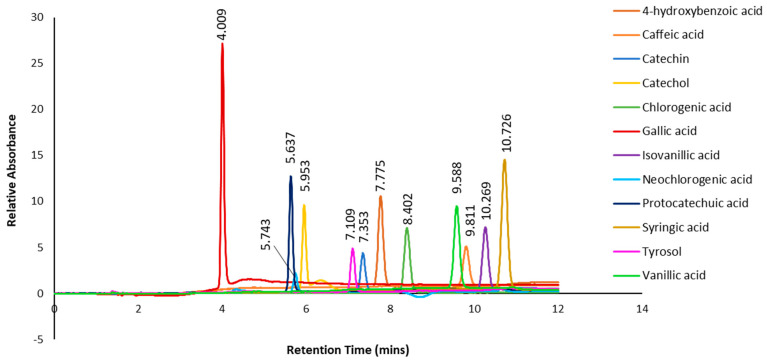
Chromatogram and retention times of selected phenolic standards.

**Figure 8 plants-13-00807-f008:**
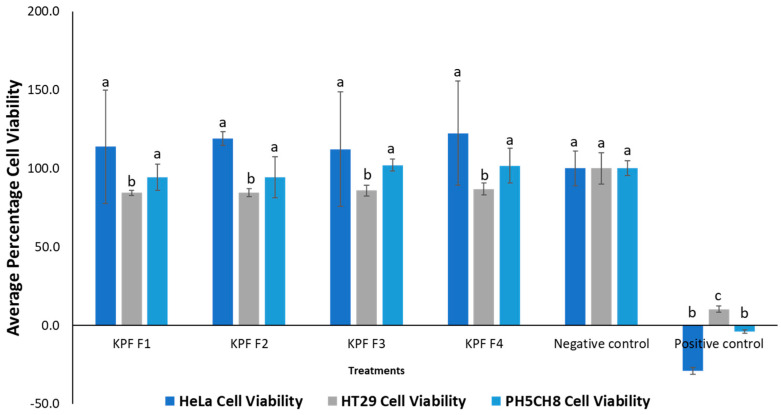
Percentage cell viability of cell lines treated with *Terminalia ferdinandiana* fractions. One-way ANOVA test indicated no significant difference (*p* > 0.05) in cytotoxicity between the different subfractions for the same cell line, denoted by the same letters on the respective bars, except in the case of HT29 cells. Negative control: cells without treatment, positive control: cell treated with 50 ug mL^−1^ cisplatin (chemotherapy drug).

**Figure 9 plants-13-00807-f009:**
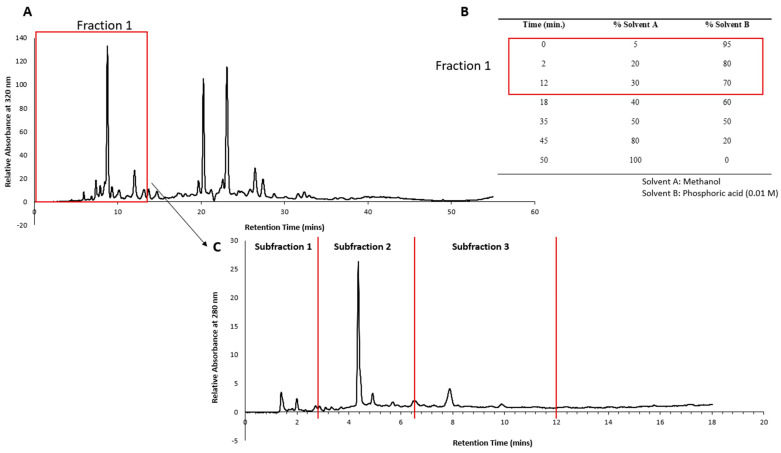
(**A**) Chromatogram of *Pittosporum angustifolium* extract showing retention times at which Fraction 1 was collected. (**B**) Elution gradient of *P. angustifolium* extract fractionation. (**C**) HPLC chromatogram of *P. angustifolium* Fraction 1 and the retention times (0–3 min (Subfraction 1), 3–6 min (Subfraction 2) and 6–11 min (Subfraction 3)) at which the three subfractions were collected.

**Figure 10 plants-13-00807-f010:**
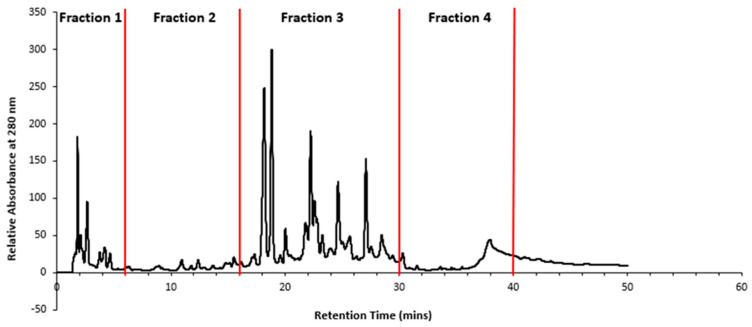
HPLC chromatogram of *T. ferdinandiana* extract and the retention times (0–6 min (Fraction 1), 6–16 min (Fraction 2), 16–30 min (Fraction 3) and 30–40 min (Fraction 4)) at which the four fractions were collected.

**Table 1 plants-13-00807-t001:** Tentative LC-MS characterization of compounds in *Pittosporum angustifolium* Fraction 1.

Peak No.	Proposed Compound	Molecular Formula	RT (min)	Mode of Ionisation	Molecular Weight (g/mol)	Observed Precursor Mass (*m*/*z*)	Theoretical Mass (*m*/*z*)	Product Ions (*m/z*)	Literature
1	Chlorogenic acid	C_16_H_18_O_9_	9.13	positive	354.31	355.10	355.00 *	65.0, 188.0, 219.0, 275.8	
1	Neochlorogenic acid	C_16_H_18_O_9_	9.13	positive	354.31	354.31	355.00	65.0, 188.0, 219.0, 275.8	Li et al., 2016 [15]
2	Unidentified	Unidentified	23.29	negative	Unidentified	229.10	Unidentified	157.1, 102.2	
3	Unidentified	Unidentified	29.15	positive	Unidentified	313.10	Unidentified	223.1, 158.2, 102.2	
4	Berberine	C_20_H_18_NO_4_^+^	30.65	positive	336.4	336.20	336.12 **	287.2	Jiao et al., 2018 [16]
5	Unidentified	Unidentified	33.09	positive	Unidentified	378.2	Unidentified	102.1, 249.2	
6	Unidentified	Unidentified	34.11	positive	Unidentified	326.9	Unidentified	102.3, 185.1, 228.3	
7	Bergapten	C_12_H_8_O_4_	36.48	positive	216.042	217.10	217.05 *	129.0, 202.0	
8	Rosmarinic acid	C_18_H_16_O_8_	44.44	positive	360.3	361.20	361.09 **	181.05, 139.04	
9	8′-epitanegool	C_20_H_24_O_7_Na	45.07	positive	399.39	399.2	399.14	287.3, 304.2	Jiao et al., 2018 [16]
10	Unidentified	Unidentified	46.07	positive	Unidentified	403.2	Unidentified	102.2, 329.2, 361	

* NIST Library. ** PubChem.

**Table 2 plants-13-00807-t002:** Target GC-MS/MS peak area and classification of compounds identified in *Pittosporum angustifolium* Fraction 1 and Fraction 1 Subfractions 1, 2 and 3.

	Target	Fraction 1	Subfraction 1	Subfraction 2	Subfraction 3	Compound Class
1	Citric acid-4TMS	2,971,610	87,281	124,060	23,465	Carboxylic acids
2	Glucose-meto-5TMS(1)	2,162,365	230,045	0	163,277	Carbohydrate
3	Inositol-6TMS(2)	1,194,216	141,218	1123	10,389	Carbocyclic sugar
4	2-Aminopimelic acid-3TMS	1,188,393	35,263	46,722	10,226	Amino acid
5	Hydroxylamine-3TMS	609,044	0	574,334	0	Hydroxylamine
6	Glucose-meto-5TMS(2)	423,257	49,736	3456	34,808	Carbohydrate
7	Galactose-meto-5TMS(2)	422,411	49,066	3587	38,804	Carbohydrate
8	1,5-13C2-Citric acid	412,327	13,144	16,485	3281	Carboxylic acids
9	Lactic acid-2TMS	268,289	437,458	247,071	444,661	Carboxylic acids
10	1,6-Anhydroglucose-3TMS	228,502	14,470	1080	4143	Carbohydrate
11	Alanine-2TMS	225,866	32,076	8606	14,237	Amino acid
12	Xylose-meto-4TMS(1)	178,497	31,305	4775	22,749	Carbohydrates
13	3-Aminoglutaric acid-2TMS	169,217	26,046	4177	2165	Amino acid
14	4-Aminobutyric acid-3TMS	169,048	17,510	0	6009	Amino acid
15	Arabinose-meto-4TMS	165,445	30,249	4666	19,117	Carbohydrates
16	Palmitic acid-TMS	154,588	129,131	126,973	120,981	Saturated fatty acid
17	Sarcosine-d3-2TMS	147,242	130,615	123,520	94,609	Amino acid
18	Lyxose-meto-4TMS(2)	146,571	23,892	3754	16,430	Carbohydrates
19	Proline-2TMS	144,359	37,171	3022	0	Amino acid
20	Malic acid-3TMS	135,932	12,652	2318	3148	Carboxylic acids

**Table 3 plants-13-00807-t003:** *Terminalia ferdinandiana* flesh (KPF) lyophilized fractions.

Fractions	Crystal Product Obtained (mg)	Concentrations of Fractions Tested (mg/mL)
KPF1	38.10	0.095
KPF 2	169.00	0.423
KPF 3	77.40	0.194
KPF 4	17.60	0.044

**Table 4 plants-13-00807-t004:** Average zone of inhibition (*n* = 3) of *Terminalia ferdinandiana* flesh fractions.

Kakadu Plum Fractions	Concentrations (µg/mL)	Bacterial Strain Zone of Inhibition (mm)			
		Gram Positive	Gram Negative		
		*S. aureus*	*E. coli*	*P. aeruginosa*	*S. typhimurium*
1	58.1	4.00 ± 0.71	0.00	0.00	0.00
2	169.0	4.30 ± 0.35	3.20 ± 0.10	2.10 ± 0.20	2.30 ± 0.40
3	77.9	0.00	0.00	0.00	0.00
4	17.6	4.20 ± 0.25	0.00	0.00	0.00
Positive control (gentamicin)	10	13.67 ± 0.58	16.33 ± 0.58	13.00 ± 0.10	12.67 ± 0.58

All values are given as means ± 2 SD (*n* = 3).

## Data Availability

Data are contained within the article.

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
