# Peer review of "Bioassay-Guided Fractionation of Pittosporum angustifolium and Terminalia ferdinandiana with Liquid Chromatography Mass Spectroscopy and Gas Chromatography Mass Spectroscopy Exploratory Study"

_plants, 2024, doi:10.3390/plants13060807_

Round 1

Reviewer 1 Report

Comments and Suggestions for Authors

This manuscript contains many problems which I noted below.  Author should revise them and then submit the revised version to any adequate journal. 

1) The data and conclusion reported in this manuscript have already been reported in rference 19 [Mani et al., (2022). Bioassay Guided Fractionation Protocol for Determining Novel Active Compounds in Selected Australian Flora. Plants. 11(21), 2826]. 

2) The title should mean the "Bioassay guided fractionation" with a LC-MS/MS and GC-MS exploratory study.  However, in the fractionation process, LC-MS/MS and GC-MS have not been used. 

3) LC-MS/MS and GC-MS data is used for identifying the compounds.  But they are tentatively identified and some compounds have not been identikfied. 

4) The manuscript reporting only fractionation and its activity has very low importance, since fractionation process would have low reproducibility.  Same results could not be obtained by other researchers

5) In References section, references are indicated with number.  But in the body text, references are indicated with author name and published year but not indicated the reference number.  

Author Response

Word Document uploaded

Reviewer 2 Report

Comments and Suggestions for Authors

I haven't studied and reviewed such an excellent article in a long time. The way of approaching the subject is very scientific. I would not change a single step. Everything is explained in the article accurately and clearly. Everything is documented in a precise scientific way. I must congratulate the authors for the comprehensive work they have done and obviously recommend that the article be published as it stands.

Author Response

We thank reviewer 2 for his/her comments. 

Reviewer 3 Report

Comments and Suggestions for Authors

The aim of this paper is a continuation of a previous study to identify compounds responsible for the cytotoxic properties of two plant species.

Although the paper is generally well-written it is very long and the results rather inconclusive. It is recommended to find a more concise way of presenting and organizing the data. There are 17 Figures and 6 Tables. It is suggested that they are combined, or some omitted. Particularly Figs 10-14. The complete HPLC chromatogram of fraction 1 should be presented showing all peaks, and the mass spectrum data could be presented in tabular form. All of these figures are not well presented.

Why is the data for the sub-fractions presented before those of Fraction 1?

Why do the subfractions have similar composition in the HPLC chromatograms?

Why is fraction 1 not used in the cytotoxicity study for comparison?

It is not clear why the T. ferdinandiana results are included in this paper as extracts were not evaluated using LC-MS/MS and GC-MS. This is misleading also in the title.

The conclusions suggest that synergy could be responsible for the presence of toxicity. Were the recombined subfractions tested? or maybe the bioactive compound was not isolated in these subfractions? There is a lot of emphasis on retesting/confirming the results making it rather inconclusive and perhaps indicating a further paper on the same topic.

Some points are covered more specifically below:

Title

 Bioassay guided fractionation of Pittosporum angustifolium and Terminalia ferdinandiana: a LC-MS/MS and GC-MS exploratory study

“Fractionation of methanolic extracts of”

T. ferdinandiana extracts were not evaluated using LC-MS/MS and GC-MS

Abstract covers the content of the paper.

Keywords are missing.

Introduction

There is no information about the two plant species. Which part of the plant was extracted?

Does not lay out the objectives of the study.

(2, 26–29) does this refer to references?

High performance liquid chromatography (HPLC) use abbreviation at first mention and then use in text thereafter.

Page 3 on the fractions were also included. On these fractions

Figure 1. Bioassay guided fractionation protocol design (Mani, et al., 2022). There are abbreviations in the figure these should be explained in the legend. Is phase 3 a new protocol in this paper? What are plant samples A, B and C?

(Mani, et al., 2022). This reference has unnecessary, comma after authors name.

Results Discussion

2.2 “a run time of mins with post run time of 5 mins” number missing?

Figure 2. (A) Chromatogram of Pittosporum angustifolium extract showing retention times at which the five fractions were collected. (B) Elution gradient of P. angustifolium extract fractionation. (C) Chromatogram of P. angustifolium Fraction 1.

Specify: HPLC chromatogram, species name in italics, the solvents A and B used in elution gradient, retention times at which the five fractions were collected are not shown.

Figure 3. HPLC chromatogram of Pittosporum angustifolium Fraction 1 and the retention times (0-3 mins (Sub-fraction 1), 3-6 mins (Sub-fraction 2), and 6-11 mins (Sub-fraction 3)) at which the three subfractions fractions were collected. This figure is unnecessary. The information in Figure 3 would be better combined in Figure 2 C.

“was followed” were

“The gradient elution described in Figure 2 (B)” give the solvents used.

T. ferdinandiana  in italics

“human epithelial cell”  carcinoma?

3.5 hrs - use h

Fig 6 Pittosporum. Angustifolium, 50 ug mL-1.  Correct

Figure 7. HPLC profiles of Pittosporum angustifolium subfractions 1 (A), subfraction 2 (B) and subfraction 3 (C). How is it possible that that these three profiles all have peaks from 0 – 6 min. aren’t they based on time slicing?

“However, there were some discrepancies between the retention time of the standard gallic acid (4.009 mins) and the subfraction (4.412 mins).” Couldn’t this be tested by spiking the subfraction with gallic acid? Were similar concentrations used?

As the subfractions of P. angustifolium did not show any significant cytotoxic activity, further separation and isolation was deemed impractical and hence were not pursued. ?

Figure 8. Chromatogram and retention times of selected phenolic standards. Specify HPLC chromatogram. The relative absorbance of the standards is much higher than the peaks in the subfractions. Is this affecting the retention time?

3.1.2. LC- MS Analysis of P. angustifolium Fraction 1 species name in italics also Fig 9

“it was also safer in terms of its selectivity index” what does this mean?

Table 3: Tentative LC-MS characterization of compounds in Pittosporum angustifolium Fraction 1. This should be presented before Figure 9. Table is not well-aligned.

Figure 9. “Proposed” phenolic metabolites in Pittosporum angustifolium Fraction 1. Were they identified in fraction 1 or not?

work have suggested has

Figure 10. Chromatogram and mass spectrum of compound 1 in Pittosporum angustifolium Fraction 1. Figure appears distorted. All of these figures are distorted.

“plants such as Berberis vulgaris, B. aristotle, B. aquifolium, Hydrastus canadensis, Pellodendron chenins, and Coptidis rhizomes” italics

‘“speculate” the occurrence’ use different word.

Figure 11 appears distorted. The base line has a lot of noise.

“If the identities of these compounds are confirmed in Fraction 1” Here there is speculation as to the validity of the results?

Figure 15 and Table 4 appear to present the same data. One concise way would be sufficient.

Table 6 explain Kakadu plum. also in Methods and introduction.

Why wasn’t the crude extract tested together with the subfractions?

“(inositol isomer) and its Janice Maniderivatives in particular”?

Conclusions

Authors should conclude whether the aims of the paper have been achieved and comment on the validity of the results presented.

References

Correct species name to italics in all reference titles

Reference 20 and 21 are the same.

Author Response

Word document addressing the comments is uploaded. 

Round 2

Reviewer 1 Report

Comments and Suggestions for Authors

I think that the problems have been revised, and thus, this manuscript has reached to acceptable level. 

Author Response

We thank the reviewer for his comments.

Reviewer 3 Report

Comments and Suggestions for Authors

Although the authors have made extensive improvements to their manuscript the points below still remain regarding their research programming and design.

Point 1 The title is still misleading.

Reviewer. It is not clear why the T. ferdinandiana results are included in this paper as extracts were not evaluated using LC-MS/MS and GC-MS. This is misleading also in the title.

Authors. The title is now revised to:

“Bioassay guided fractionation of Pittosporum angustifolium and Terminalia ferdinandiana with LC-MS/MS and GC-MS exploratory study”.

Point 2. It is not valid to publish results which have not been rigorously or completely researched. Giving reasons of lack of time or finances are not valid. Wait till these resources are available to complete the study.

This comment by the authors refers to Point 1 and 2.

 “Since T. ferdinandiana lyophilised extracts had shown some promising activity in our previous paper and since it is an understudied species, fractionation of the extracts was worthwhile. However, due to time limitations LC-MS/MS or GC-MS studies were not performed, although it would be ideal for future work.”

“Due to funding and time limitations these retests and confirmatory studies could not be performed.”

Point 3 There is some question raised by the authors as to the reproducibility of their results.

Reviewer. Conclusions: Authors should conclude whether the aims of the paper have been achieved and comment on the validity of the results presented.

Authors. The conclusion has now been revised.

Overall, this study successfully applied the third phase of our developed bioassay guided fractionation protocol (Figure 1). However, further studies and retests are warranted to confirm and check for the reproducibility of the results obtained.

Author Response

We thank reviewer 3 for the feedback. 

The authors would like to reiterate that this was merely a preliminary study to assess the scope of the proposed bioassay guided fractionation protocol. As such it was exploratory in nature and the aims are to extend on initial research in the future. 

Point 1 and 2

Although LC-MS/MS and GC-MS studies were not performed on T. ferdinandiana, it was pertinent to include these results to demonstrate the continuity of our proposed bioassay guided fractionation protocol and findings from our previous study. T. ferdinandiana also presented significant bioactivity and its inclusion in this study will inform researchers of potential research directions for this understudied species.  

The authors believe that the revised title “Bioassay guided fractionation of Pittosporum angustifolium and Terminalia ferdinandiana with LC-MS/MS and GC-MS exploratory study” means that the study demonstrate the bioassay guided fractionation protocol of the two plants and includes LC-MS/MS and GC-MS exploratory study. The word "with" implies that LC-MS/MS and GC-MS are included, but not necessarily specifying that it has been performed on both species. Moreover, the LC-MS/MS and GC-MS studies were only exploratory in nature which is highlighted in the titile. 

Point 3

As mentioned earlier this was a pilot study as such several limitations of the study were presented and have been highlighted in the paper. The purpose of this study was to design a quick and robust bioassay guided fractionation protocol to be able to screen several understudied native Australian plants. Although, our work future studies need further improvement in testing reproducibility and validation, this study has been successful in developing a systematic approach in the discovery of novel medicinal plants and fills the knowledge gap of understudies species. 

Round 3

Reviewer 3 Report

Comments and Suggestions for Authors

I am willing to accept the argument for the title.

Reading the authors comments to the one of the other reviewers it became evident that the authors do not have a library for their MS data and this why they are so tentative and inclusive about their results.

I recommend that in more detailed future work that the use of an MS library would be necessary.